# Triterpenes and Phenolic Compounds from *Euphorbia deightonii* with Antiviral Activity against Herpes Simplex Virus Type-2

**DOI:** 10.3390/plants11060764

**Published:** 2022-03-13

**Authors:** Muhammad Bello Saidu, Norbert Kúsz, Yu-Chi Tsai, Máté Vágvölgyi, Róbert Berkecz, Dávid Kókai, Katalin Burián, Judit Hohmann, Dóra Rédei

**Affiliations:** 1Department of Pharmacognosy, University of Szeged, Eötvös u. 6, 6720 Szeged, Hungary; bello.saidu@pharmacognosy.hu (M.B.S.); kusznorbert@gmail.com (N.K.); yuchi0713@gmail.com (Y.-C.T.); vagvolgyi.mate@gmail.com (M.V.); redei.dora.judit@szte.hu (D.R.); 2Institute of Pharmaceutical Analysis, University of Szeged, Somogyi u. 4, 6720 Szeged, Hungary; berkecz.robert@szte.hu; 3Department of Medical Microbiology, Albert Szent-Györgyi Medical School, University of Szeged, Dóm tér 10, 6720 Szeged, Hungary; kokai.david@med.u-szeged.hu (D.K.); burian.katalin@med.u-szeged.hu (K.B.); 4Interdisciplinary Centre of Natural Products, University of Szeged, Eötvös u. 6, 6720 Szeged, Hungary

**Keywords:** *Euphorbia deightonii*, Euphorbiaceae, triterpenes, neolignans, coumarin, trimethyl-ellagic acid

## Abstract

Two undescribed compounds, 3*β,*7*β*-dihydroxy-24-methylenelanosta-8-ene-11-one (**1**) and neolignane deightonin (**4**) were isolated from the aerial parts of *Euphorbia deightonii* Croizat together with six known compounds, namely, kansenone (**2**), euphorbol-7-one (**3**), dehydrodiconiferyl diacetate (**5**), marylaurencinol D (**6**), scoparon (**7**), and 3,4,3′-tri-*O*-methylellagic acid (**8**). The structures of the isolated compounds were determined by HRESIMS, 1D (^1^H, ^13^C JMOD) and 2D NMR (HSQC, HMBC, ^1^H–^1^H COSY, NOESY) spectroscopic analysis, and by comparison of the assignments with literature data. The anti-herpes simplex virus type-2 activity of the isolated compounds were investigated by qRT-PCR assay on Vero cells after determining cytotoxic concentration 50% (CC_50_). Compounds **1**, **3**, **4**, and **7** exhibited inhibitory effects with respective IC_50_ values of 7.05, 2.42, 11.73, and 0.032 µM. Scoparon (**7**) showed the strongest anti-HSV activity with a selectivity index of 10.93.

## 1. Introduction

The herpes simplex virus (HSV) causes one of the most common viral infections in humans, in whom infection causes a variety of disorders ranging from mild to severe diseases, and in certain cases life-threatening illness. Two different types of HSV are distinguished: HSV-1 and HSV-2; the former is primarily responsible for oral herpes cases, while the latter causes genital herpes. It has been estimated that more than two-thirds of the population under the age of 50 is infected with either HSV-1 or HSV-2 (*cc*. 67% and 13%, respectively) [1]. Eradication of both HSV-1 and HSV-2 from the body is not possible, as they cause lifelong infections. Although most of the herpes infections are asymptomatic, some may experience painful and blistering sores, as well as flu-like symptoms, such as enlarged lymph nodes, tiredness, headache, etc. during periods of exacerbation. An effective anti-HSV vaccine is yet to be developed; however, in the meantime, two medications—acyclovir and valacyclovir—are capable of alleviating the symptoms of the infection [2].

Many plants, including *Euphorbia* species, are used worldwide in traditional medicine to treat HSV infections. In a study, 47 plant extracts of 10 *Euphorbia* species used by Colombian traditional therapy were tested in vitro for their potential antiherpetic activity. Among the tested extracts, five exhibited an antiherpetic effect representing three *Euphorbia* species. The highest activity was found in the extracts of *E. cotinifolia* and *E. tirucalli* [3].

Regarding the active metabolites of Euphorbia extracts, different metabolites were found with anti-HSV activity. In case of *E. thymifolia,* the ethyl acetate extract and its constituent, 3-*O*-galloyl-4,6-(*S*)-hexahydroxydiphenoyl-D-glucose, have been shown to inhibit HSV-2 multiplication by reducing HSV-2 infectivity [4]. The tannin-type metabolite of *E. jolkini*, putranjivain A, was found to significantly reduce viral infectivity. This compound also showed inhibition of viral attachment and cell penetration. The combination of putranjivain A and acyclovir produced no interaction [5]. In case of *E. segetalis,* 11 triterpenes were evaluated for their antiviral activities against HSV, and lupenone exhibited a strong viral plaque inhibitory effect against HSV-1 and HSV-2 [6].

The present study deals with the investigation of *Euphorbia deightonii* Croizat. This plant, which is a latex-rich, cactus-like shrub growing up to 6 m high, is indigenous to West Africa. Historically, it has been used as an ornamental plant, while its preparations were used against leprosy and as arrow poison during tribal wars [7]. The present paper describes the isolation and structure elucidation of two novel (one triterpene and one neolignane) and six known compounds from *E. deightonii*, together with the investigation of their anti-HSV-2 activity.

## 2. Results and Discussion

### 2.1. Isolation and Structure Elucidation of Compounds

Large quantity of the fresh aerial parts of *E. deightonii* was collected from Zaria city, Kaduna State of Nigeria. The harvested plant material was washed to remove soil debris, then air dried, cut into smaller pieces, and ground into powdered form. Finally, 1.2 kg of the air-dried powdered sample was exhaustively extracted with methanol using percolation method at room temperature. The methanol extract was evaporated until dry, re-dissolved in MeOH–H_2_O (1:1), and extracted with chloroform. The chloroform-soluble phase was fractionated by open column chromatography on a polyamide stationary phase with different ratios of MeOH-H_2_O as eluents. The fraction obtained with MeOH-H_2_O (3:2) was further chromatographed by vacuum liquid chromatography (VLC) on normal- and reversed-phase silica gel, and subsequently by NP- and RP-HPLC, and preparative TLC to isolate eight pure compounds. Structure elucidation was performed by means of HRESIMS, and 1D (^1^H, ^13^C JMOD) and 2D NMR (HMBC, HSQC, ^1^H–^1^H COSY, NOESY) spectroscopic analysis.

Compound **1** was isolated as a white amorphous powder. The protonated molecular ion at *m/z* 471.3844 [M + H]^+^ (calcd for 471.3833, C_31_H_51_O_3_) in the HRESIMS spectrum provided the molecular formula C_31_H_50_O_3_. The ^1^H NMR spectrum displayed the resonances of five tertiary (*δ*_H_ 0.85, 0.92, 1.01, 1.05, and 1.17, each 3H, s) and three secondary (*δ*_H_ 0.92, d, *J* = 6.5 Hz; *δ*_H_ 1.02 and 1.03, each d, *J* = 6.7 Hz, 3H) methyl groups, two oxymethines (*δ*_H_ 3.32, dd, *J* = 11.2 and 5.0 Hz; *δ*_H_ 4.27, br s), and an exomethylene (*δ*_H_ 4.73, br s; *δ*_H_ 4.66, d, *J* = 1.2 Hz) (Table 1). The presence of a tetrasubstituted olefin bond was apparent from the nonprotonated *sp*^2^ carbon signals at *δ*_C_ 141.6 and 157.3. The 1D NMR data suggested that compound **1** is a tetracyclic triterpene with an exomethylene-containing side chain, similar to that of euphorbol-7-one (**3**) and its derivatives previously isolated from *Polyalthia oblique* [8]. The planar structure of **1** was assembled with the aid of HSQC, ^1^H–^1^H COSY, and HMBC data. It was found that two hydroxy-groups are located at C-3 (*δ*_H_ 3.32, *δ*_C_ 78.8) and C-7 (*δ*_H_ 4.27, *δ*_C_ 65.6), while a keto substituent was placed onto C-11 (*δ*_C_ 200.4), which formed a conjugated enone with a Δ^(8,9)^ double bond. The relative configuration of **1** was established based on relevant NOE correlations and coupling constant values. The large *J* value of H-3 (11.2 Hz) dictated that 3-OH is oriented equatorially, thus occupying a β-position [9]. This conclusion was in agreement with NOE correlations between H-3, H-5, and H_3_-29. Considering a strong NOE between H-7 and H_3_-30, the 7-hydroxy group must be in an β-position. Furthermore, in the H-17α/H_3_-21, H-20/H_3_-18, and H-12α/H_3_-21 NOESY correlations, the chemical shift of H_3_-21 (*δ*_H_ 0.92, d, *J* = 6.5 Hz) was consistent with literature data reported for lanostane-type triterpenes [8]. Accordingly, compound **1** was identified as 3*β,*7*β*-dihydroxy-24-methylenelanosta-8-ene-11-one.

Compound **4** was obtained as a colourless oil. The molecular formula C_23_H_26_O_7_ was assigned for **4** based on its molecular ion peak at *m/z* 415.1757 [M+H]^+^ (calcd for 415.1751, C_23_H_27_O_7_) in the HRESIMS spectrum. Its ^1^H NMR spectrum showed characteristic signals of a neolignane, including a 1,3,4-trisubstituted benzene and a dihydrobenzofurane ring (see Table 2). Comparison of the 1D NMR data with literature data revealed that compound **4** is similar to dihydrocarinatinol previously reported from *Virola carinata* [10]. However, unlike in dihydrocarinatinol, the allyl side chain of **4** is substituted with a methoxy group, as shown by the HMBC correlations of C-7’ (*δ*_C_ 84.8) with H-9’ (*δ*_H_ 5.23 and 5.29) and 7′-OCH_3_ (*δ*_H_ 3.34), while the hydroxymethyl moiety is acetylated, as suggested by heteronuclear correlations between the acetyl carbonyl (*δ*_C_ 170.9) and H_2_-9 oxymethylene protons (*δ*_H_ 4.29 and 4.47). The vicinal coupling constant value of 7.6 Hz between H-7 and H-8 dictated their *trans* relationship [11]. Trivial name deightonin was given for **4**; its chemical name is *trans*-(2-(4-hydroxy-3-methoxyphenyl)-7-methoxy-5-(1-methoxyallyl)-2,3-dihydrobenzofuran-3-yl)methyl acetate. Although the absolute configuration has not been determined, compound **4** is most likely a mixture of C-7′ epimers, as indicated by the duplicated proton and carbon signals at and around C-7′. Chromatography of **4** on chiral HPLC column afforded 4 peaks in an approximately 1:1:5:5 area% ratio, indicating four stereoisomers (see Appendix A). 7*S*,8*R* or 7*R*,8*S* of *trans*-oriented H-7/H-8, and *R*/*S* configuration of C-7′.

The known compounds were identified based on the literature data as kansenone (**2**) [9], euphorbol-7-one (**3**) [12], dehydrodiconiferyl-diacetate (**5**) [13], marylaurencinol D (**6**) [14], scoparon (=6,7-dimethoxycoumarin, syn. esculetin) (**7**) [15], and 3,4,3′-tri-*O*-methylellagic acid (**8**) [16].

### 2.2. Cytotoxicity and Anti-HSV-2 Activity of the Compounds

First, the CC_50_ concentration of the compounds (**1**–**8**) Figure 1, (Table 3) was determined by MTT assay on Vero cells; then the possible antiviral effect of the substances was measured. As shown on Figure 2, 3*β,*7*β*-dihydroxy-24-methylenelanosta-8-ene-11-one (**1**) (Figure 2a), euphorbol-7-one (**3**) (Figure 2b), deightonin (**4**) (Figure 2c), and scoparon (**7**) (Figure 2d) exhibited an anti-HSV-2 effect. Furthermore, IC_50_ values of compounds **1**, **3**, **4**, and **7** were 7.05, 2.42, 11.73, and 0.032 µM, respectively. Among the tested triterpenes, the lanostane (**1**) and 24-methylenelanostane (**3**) proved to be effective, while among the neolignans, only deightonin (**4**) with the methoxylated allyl group showed an antiviral effect. The most pronounced activity was exerted by the coumarin scoparon (**7**); its effect (IC_50_ 32.05 nM!) was comparable to that of acyclovir in the respective concentration.

Previous studies have demonstrated that kansenone (**2**) exhibits strong cytotoxicity against both rat epithelioid [17] and human normal liver (L–O2) and gastric epithelial cell lines (GES–2) [18]. It was found that kansenone (**2**) could up-regulate the apoptotic proteins Bax, AIF, Apaf-1, cytochrome c, caspase-3, caspase-9, caspase-8, FasR, FasL, NF-κB, and TNFR1 mRNA expression levels, and down-regulate the anti-apoptotic Bcl-2 family proteins, revealing its apoptosis-inducing activity through both the death receptor and mitochondrial pathways [17]. No antiviral activity was reported previously for kansenone (**2**). Euphorbol-7-one (**3**) was tested against six terrestrial pathogenic bacteria but proved to be ineffective [8]. Dehydrodiconiferyl-diacetate (**5**) was isolated previously from two species of *Euphorbia* genus [19,20]. It displayed moderate cytotoxic activity against human HT-1080 fibrosarcoma and murine colon 26-L5 carcinoma cells [21]. Marylaurencinol D (**6**) was obtained before from an Orchidaceae species and isolated in our experiment for the first time from an *Euphorbia* species. This compound was investigated by Yoshikawa et al. for its antimicrobial activity and showed weak potency against *Bacillus subtilis* and had no effect against *Klebsiella pneumoniae* and *Trychophyton rubrum* [15]. Scoparon (**7**) is the only compound that has been studied for antiviral activity. Its anti-hepatitis B virus (HBV) activity was investigated in vitro and in vivo. The results provided evidence that this compound efficiently inhibits viral replication in a human hepatocellular liver carcinoma 2.2.15 (HepG2.2.15) model transfected with HBV [22]. However, scoparon (**7**) was not effective against influenza A virus (A/PR/8/34, H1N1) (IC_50_ > 15 µg/mL) [23]. In another study, scoparon (**7**) exerted porcine circovirus type 2 (PCV2) replication-inducing activity at the concentration of 0.01563 mg/mL, suggesting that it can facilitate PCV2 growth in cell culture; its mechanism of action is not known [24].

Ellagic acid is widely spread among higher plants; it is predominately found in ester-linked to sugars in hydrolysable tannins called ellagitannins. In methylated form, ellagic acid (such **8**) is much less common. The antiviral activity of ellagitannins (castalagin, vescalagin, and grandinin) was previously analysed and activity against acyclovir (ACV)-resistant strains of HSV-1 and HSV-2 was proved [25], but no antiviral effect was reported for methylated derivatives.

## 3. Materials and Methods

### 3.1. Plant Material

The aerial parts of *E. deightonii* were obtained in August 2018 from Zaria, Kaduna State of Nigeria (at latitude 11°5′4.2252″ N and longitude 7°45′44.172″ E). The plant material was identified by Umar Gallah (Biological Department, Ahmadu Bello University, Zaria, Nigeria). A voucher specimen (No. 0918) has been deposited in the Herbarium of Department of Pharmacognosy, University of Szeged, Szeged, Hungary.

### 3.2. General Experimental Procedures

Optical rotations were measured with a Jasco P-2000 polarimeter (Jasco International Co. Ltd., Hachioji, Tokyo, Japan). NMR spectra were recorded in CDCl_3_ on a Bruker Avance DRX 500 spectrometer (Bruker, Billerica, MA, USA) at 500 MHz (^1^H) and 125 MHz (^13^C). The signals of the deuterated solvents were taken as references. Two-dimensional NMR measurements were carried out with standard Bruker software. In the COSY, HSQC, and HMBC experiments, gradient-enhanced versions were applied. The HRMS were acquired on a Thermo Scientific Q-Exactive Plus Orbitrap mass spectrometer equipped with an ESI ion source in positive ionization mode. The resolution was over 1 ppm. The data were acquired and processed with MassLynx software.

Vacuum-liquid chromatography (VLC) was carried out on silica gel (15 μm, Merck, Darmstadt, Germany); LiChroprep RP-18 (40–63 μm, Merck, Darmstadt, Germany) stationary phase was used for reversed-phase VLC; column chromatography (CC) was performed on polyamide (MP Biomedicals, Santa Ana, CA, USA). Preparative thin-layer chromatography (prep. TLC) was performed on silica gel 60 F_254_ plates (Merck, Darmstadt, Germany). HPLC was carried out on a Waters HPLC instrument (Waters, Milford, MA, USA), using normal Phenomenex Luna Silica (3 μm 100 A) and reversed-phase [Phenomenex, Kinetex 5 μm C18 100A, and LiChrospher LiChroCART 250-4 RP-18e (5 μm)] columns (Phenomenex, Torrance, CA, USA). Chiral chromatographic separation of compound **4** was performed on a Jasco HPLC/SFC instrument (Jasco International Co. Ltd., Hachioji, Tokyo, Japan) using a Phenomenex Lux^®^ 5 µm i-Amylose-1, 250 × 4.6 mm LC column (Phenomenex, Inc., Torrance, CA, USA). The instrument was equipped with an MD-4015 photodiode array detector collecting data in a detection range of 210–410 nm. The mobile phase eluent was an 87:13 ratio of cyclohexane and tetrahydrofuran with 0.1% TFA and applied at a constant 1 mL/min flow rate.

### 3.3. Extraction and Isolation

The harvested plant material was washed in order to remove soil debris, then air-dried, cut into smaller pieces, and ground into a powdered form (1200 g). The powdered material was exhaustively percolated with MeOH (29 L) at room temperature. The crude extract (218 g) was concentrated under reduced pressure, resuspended in aqueous methanol, and then solvent–solvent partitioning was carried out with CHCl_3_ (6 × 500 mL). After concentration, the organic phase (134 g) was subjected to open polyamide column chromatography using gradient mixtures of MeOH–H_2_O (1:4, 3:2, 4:1, and 5:0) as eluents. The fraction (21.1 g) eluted with MeOH–H_2_O (3:2) was fractionated by vacuum liquid chromatography (VLC) with petroleum ether–EtOAc gradient systems (9:1, 19:1, 8:2, 7:3, and 0:10) to collect 130 fractions (each 50 mL). The fractions were monitored on TLC, and those with similar chemical compositions were combined into fractions A–K. Fractions I–K obtained with petroleum ether–EtOAc (8:2 and 7:3) were re-chromatographed by NP- and RP-VLC with cyclohexane–EtOAc–EtOH and MeOH–H_2_O, respectively, to give 8 subfractions. NP-VLC of fraction I with cyclohexane–EtOAc–EtOH gave fraction I/I, which was further separated by RP-VLC with MeOH–H_2_O eluent affording I/I/a-f subfractions. Further purification of I/I/f by RP-HPLC (MeOH–H_2_O 8:2) led to the isolation of compound **3** (5.1 mg), and after NP-PTLC using cyclohexane–CHCl_3_–EtOH (80:20:10) as mobile phase compound **2** (14.2 mg). Fraction J was subjected to NP-VLC using cyclohexane–EtOAc–EtOH as eluent (60:40:0, 80:20:1, 80:20:2, 80:20:4, 80:20:8 and 80:20:12) to collect fractions J/I and J/II. Fraction J/II was further purified by RP-VLC using MeOH–H_2_O (6:4, 7:3, 8:2, 9:1 and 10:0) eluents to obtain fractions J/II/a–e. Further purification of J/II/e by RP-HPLC resulted in fraction J/II/e/5 upon which NP-HPLC purification led to the isolation of **1** (1.6 mg). NP-VLC of fraction K gave two fractions, K/I and K/II. RP-VLC with MeOH-H_2_O solvent system on K/I gave fractions K/I/a-c while K/II gave fraction K/II/a-f. RP-HPLC conducted on K/I/a gave fraction K/I/a/2, which formed a precipitate from *n*-hexane–EtOAc mixture. This insoluble part was purified by PTLC with chloroform–acetone (19:1) as mobile phase affording the isolation of **7** (9.3 mg). On dissolution in MeOH, K/I/c gave two fractions, one insoluble in methanol (K/I/c/A) and the other soluble in methanol (K/I/c/B). Further purification of K/I/c/B using NP-HPLC (mobile phase: cyclohexane–EtOAc–MeOH 80:19:1) led to the isolation of **4** (1.4 mg), **5** (1.0 mg), and **6** (0.9 mg). Incidentally, upon observation in the solubility test, K/II/b, K/II/c, K/II/d, and K/II/f resulted in two fractions each. One set of fractions was insoluble in MeOH and CHCl_3_ but only soluble in DMSO and the other set was soluble in MeOH and CHCl_3_. TLC analysis of these methanol-insoluble fractions showed similar Rf values, colour, and visibility using UV and were combined as compound **8** (4.3 mg).

### 3.4. 3β,7β-Dihydroxy-24-methylenelanosta-8-ene-11-one (1)

White amorphous powder; [α]^24^_D_ +8.5 (*c* 0.04, CH_3_OH); ^1^H and ^13^CNMR data, see Table 1; HRESIMS positive *m/z* 471.3844 [M + H]^+^ (calcd for 471.3833, C_31_H_51_O_3_), 493.3654 [M + Na]^+^ (calcd for 493.3652, C_31_H_50_O_3_Na).

### 3.5. Deightonin (4)

Colourless oil; [α]^24^_D_ +13.5 (*c* 0.05, CH_3_OH); UV λ_max_ nm (log ε): 249 (3.83), 283 (3.80), 286 (3.79); ^1^H and ^13^C NMR data, see Table 2; HRESIMS positive: *m/z* 415.1757 [M + H]^+^ (calcd for 415.1751, C_23_H_27_O_7_), 437.1581 [M + Na]^+^ (calcd for 437.1571, C_23_H_26_O_7_Na).

### 3.6. Cultivation and Quantification of HSV-2

HSV-2 strain (donated by Dr. Ilona Mucsi, University of Szeged, Szeged, Hungary) was grown in Vero cells (ATCC) and infectivity was measured in the same cell line by using the plaque titration method [26].

### 3.7. Assay for Cytotoxic Study

For the determination of cytotoxic concentration 50% (CC_50_), a cell viability test was performed as an MTT assay using the Vero cell line. Cells were seeded at a density of 4 × 10^4^ cells/well. After an overnight period, fresh medium was complemented with serial 2-fold dilutions of all compounds **1**–**8**. The cells were incubated at 37 °C for 24 h. Later, 20 μL of thiazolyl blue tetrazolium bromide (MTT; Sigma-Aldrich, St. Louis, MO, USA) was added to each well. After an additional incubation at 37 °C for 4 h, sodium dodecyl sulphate (Sigma-Aldrich, St. Louis, MO, USA) solution (10% in 0.01 M HCI) was added and incubated overnight. CC_50_ was then determined by measuring the OD at 550 nm (ref. 630 nm) with the EZ READ 400 ELISA reader (Biochrom, Cambridge, UK). The assay was replicated four times for each concentration.

### 3.8. Assay for Anti-HSV-2 Study

The possible antiviral activity of the compounds was investigated in Vero cell lines using each compound in a non-toxic concentration. Cells were seeded in 96-well plates 4 × 10^4^ cells/well and infected at multiplicity of infection (MOI) of 0.01. After 1 h incubation, the cells were washed with PBS to remove the unattached HSV-2 and the PBS was replenished with a medium containing 2-fold dilutions of compounds, each starting with the first non-toxic concentration. Acyclovir was used as a positive control in the same concentration as the compounds. After 24 h incubation at 37 °C by 5% CO_2_ concentration, the cells were washed and suspended in Milli-Q water (MQ) (Millipore, Billerica, MA, USA). After 2 thaw–freeze cycles, the cells were resuspended. Using HSV-2 specific gD2 primers F: 5-TCA GCG AGG ATA ACC TGG GA-3′, R: 5-GGG AGA GCG TAC TTG-CAG GA-3, qRT-PCR was performed to check the HSV-2 genome quantity, as described earlier [27]. Measurement was performed with Bio-Rad CFX96 (Bio-Rad, Hercules, CA, USA). Selectivity index was calculated as described earlier (SI = (CC_50_/IC_50_) [28].

## 4. Conclusions

A combination of different chromatographic techniques resulted in the isolation of eight compounds; among them are three triterpene derivatives (**1**–**3**), three neolignans (**4**–**6**), a coumarin (**7**), and a trimethylated ellagic acid derivate (**8**) from the chloroform extract of the aerial parts of *E. deigtonii*. 3*β*,7*β*-Dihydroxy-24-methylenelanosta-8-ene-11-one (**1**) and deightonin (**4**) are undescribed natural products; moreover, all the isolated compounds were described for the first time from this plant species. The isolated compounds (**1**–**8**) were studied for HSV-2 inhibitory activity, and 3*β*,7*β*-dihydroxy-24-methylenelanosta-8-ene-11-one (**1**), euphorbol-7-one (**3**), deightonin (**4**), and scoparon (**7**) were found to have antiviral activities with IC_50_ values in the range of 0.032–11.73 µM. Scoparon (**7**) showed the highest selectivity index of 10.923; therefore, this compound may have an impact for future developments of antiviral drugs.

## Figures and Tables

**Figure 1 plants-11-00764-f001:**
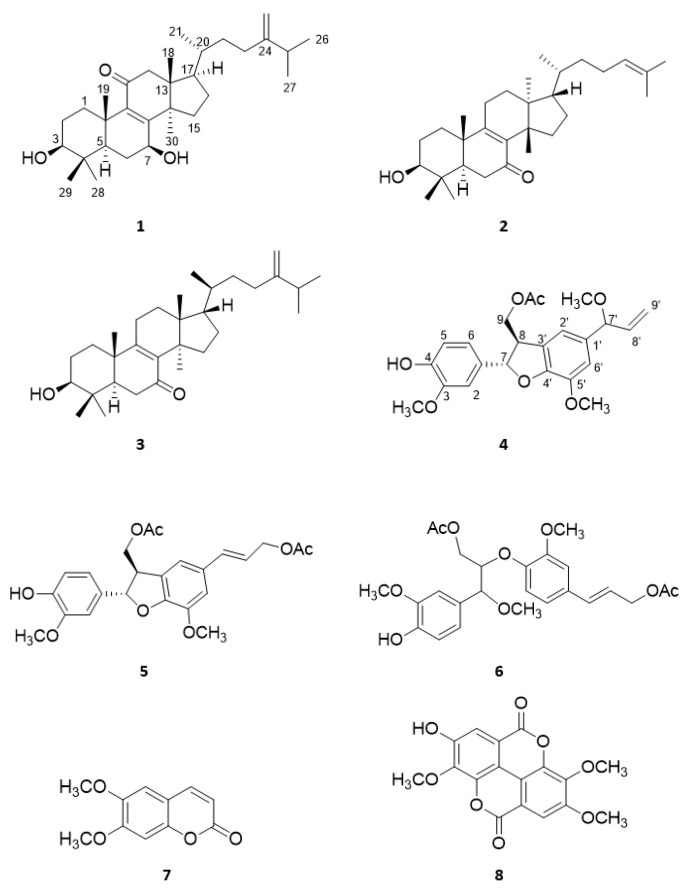
Compounds isolated from *E. deightonii* (**1**–**8**).

**Figure 2 plants-11-00764-f002:**
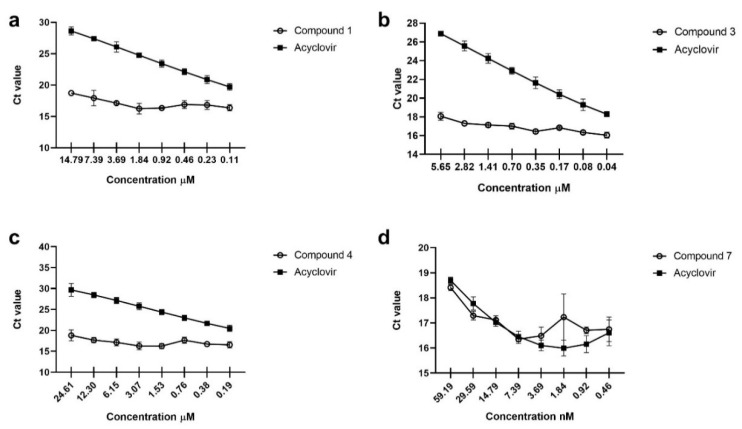
Antiviral activity of compounds. Vero cells were infected with 0.01 MOI HSV-2; after 1 h of incubation, the cells were washed and cultured with a two-fold serial dilution compound containing a medium. After a 24 h incubation period, qRT-PCR measurement using HSV-2 specific primers was carried out to check the antiviral effect of 3*β*,7*β*-dihydroxy-24-methylenelanosta-8-ene-11-one (**1**) (**a**), euphorbol-7-one (**3**) (**b**), deightonin (**4**) (**c**), and scoparon (**7**) (**d**). Acyclovir was used as a positive antiviral substance.

**Table 1 plants-11-00764-t001:** ^1^H (500 MHz) and ^13^C NMR (125 MHz) data of compound **1** in CDCl_3_ (*δ*_H_ in ppm, *J* in Hz).

Position	^1^H	^13^C
1a	1.02, m	33.9
1b	2.56, m	
2	1.66–1.72, m (2H)	28.0
3	3.32, dd (11.2, 5.0)	78.8
4	–	38.5
5	1.42, m	45.5
6a	1.69, m	28.6
6b	1.83, d (13.8)	
7	4.27, br s	65.6
8	–	157.3
9	–	141.6
10	–	38.6
11	–	200.4
12a	2.42, d (16.7)	51.6
12b	2.58, d (16.7)	
13	–	44.3
14	–	50.9
15a	1.39, m	29.8 *
15b	2.37, m	
16a	1.43, m	29.8 *
16b	2.05, m	
17	1.69, m	50.7
18	0.92, s (3H)	18.11
19	1.17 s (3H)	18.12
20	1.43, m	36.4
21	0.92, d (6.5) (3H)	18.5
22a	1.16, m	34.7
22b	1.58, m	
23a	1.90, m	31.3
23b	2.12, m	
24	–	156.7
25	2.23 sept (6.7)	34.0
26	1.03 *, d (6.7) (3H)	22.0
27	1.02 *, d (6.7) (3H)	22.1
28	1.05, s (3H)	28.3
29	0.85, s (3H)	16.2
30	1.01, s (3H)	25.9
1′a	4.73, br s	106.3
1′b	4.66, d (1.2)	

* overlapping signals.

**Table 2 plants-11-00764-t002:** ^1^H (500 MHz) and ^13^C NMR (125 MHz) data of compound **4** in CDCl_3_ (*δ*_H_ in ppm, *J* in Hz).

Position	^1^H	^13^C
1	–	132.7
2	6.91, m	108.9
3	–	146.9
4	–	146.0
5	6.88 *, m	114.5
6	6.88 *, m	119.76, 119.74
7	5.46, d (7.6)	88.9
8	3.78, m	50.7
9a	4.47, dd (11.2, 5.4)	65.72, 65.67
9b	4.29, dt (11.2, 7.8)	
1′	–	134.93, 134.96
2′	6.79 *, m	111.19, 112.23
3′	–	127.5
4′	–	147.8
5′	–	144.6
6′	6.79 *, m	115.27, 115.35
7′	4.56, br d (7.0)	84.76, 84.87
8′	5.93 *, ddd (17.1, 10.3, 7.0)5.92 *, ddd (17.1, 10.3, 7.0)	138.90, 138.95
9′a	5.29, dd (17.1, 1.2)	116.5
9′b	5.23, dd (10.3, 1.6)	
4-OH	5.61, s	-
3-OCH_3_	3.87, s (3H)	56.2
5′-OCH_3_	3.90, s (3H)	56.3
7′-OCH_3_	3.34, s (3H)	56.5
9-Ac Me	2.01, s (3H)	20.9
9-Ac CO	–	170.9

* overlapping signals.

**Table 3 plants-11-00764-t003:** Cytotoxic (CC_50_) and antiviral activity (IC_50_) of compounds **1**–**8**. For determination of CC_50_ concentration, Vero cells were seeded at a density of 4 × 10^4^/well. After incubating overnight, the medium was replenished with medium containing two-fold dilutions of each compound. Twenty-four hours later, MTT assay was carried out. The antiviral activity was expressed as IC_50_ in µM.

Compounds	CC_50_	Anti-HSV-2 Activity IC_50_	Selectivity Index
3*β,*7*β*-Dihydroxy-24-methylenelanosta-8-ene-11-one (**1**)	35.49 ± 1.62 µM	7.05 ± 0.25 µM	5.03
Kansenone (**2**)	52.14 ± 2.21 µM	inactive	
Euphorbol-7-one (**3**)	7.84 ± 0.96 µM	2.42 ± 0.06 µM	3.23
Deightonin (**4**)	39.76 ± 4.73 µM	11.73 ± 0.79 µM	3.389
Dehydrodiconiferyl-diacetate (**5**)	82.533 ± 8.94 µM	inactive	
Marylaurencinol D (**6**)	71.64 ± 5.83 µM	inactive	
Scoparon (**7**)	0.35 ± 0.016 µM	0.032 ± 0.0021 µM	10.923
3,4,3′-Tri-*O*-methylellagic acid (**8**)	25.74 ± 1.84 µM	inactive	
Acyclovir	100 ± 6.15 µM	0.77 ± 0.032 µM	129.87

## Data Availability

Not applicable.

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
