# Peer review of "Triterpenes and Phenolic Compounds from *Euphorbia deightonii* with Antiviral Activity against Herpes Simplex Virus Type-2"

_plants, 2022, doi:10.3390/plants11060764_

Round 1

Reviewer 1 Report

The manuscript is very interesting and it is written well.
All the experiments are done as required. I have only a few suggestions to include.
Author explained all the chromatography in the experimental section. Did the author do TCL of the crude extract after it fractionation.?

Why did the author not include the Antiviral activity of the crude extract? 

Reviewer 2 Report

In this article Saidu and co-workers described the isolation of six known compounds and two undescribed compounds from the aerial parts of Euphorbia deightonii Croizat. The authors fully characterized the new extracted compounds and then evaluated the anti-herpes simplex virus type-2 activity of the isolated compounds. The work is well written and should be published in Plants after minor revisions.
I think that the authors should report in the supplementary material:
1.    the structures of compounds 1 and 4 near the NMR spectra in order to give readable the corresponding spectra.
2.    The NMR spectra of euphorbol-7-one (3) and dihydrocarinatinol, because the authors reported that the structures of 1 and 4 have been established by the similarity with these compounds. I think that with the NMR reported it should be more easy understand what the authors suggested.
3.    The HPLC chromatograms of compounds 1 and 4 and the purity of the isolated products.

Reviewer 3 Report

Dear Authors!

I marked my point of view in yellow in the manuscript. Please verify the repetition for reference nr. 17 in the position 19. 

Best regards,

MP
